# Central Nervous System Targeted Protein Degraders

**DOI:** 10.3390/biom13081164

**Published:** 2023-07-25

**Authors:** Bedwyr ab Ion Thomas, H. Lois Lewis, D. Heulyn Jones, Simon E. Ward

**Affiliations:** 1Medicines Discovery Institute, Cardiff University, Cardiff CF10 3AT, UKlewisl33@cardiff.ac.uk (H.L.L.);; 2Chemistry Department, Cardiff University, Cardiff CF10 3AT, UK

**Keywords:** PROTACs, targeted protein degraders, CNS

## Abstract

Diseases of the central nervous system, which once occupied a large component of the pharmaceutical industry research and development portfolio, have for many years played a smaller part in major pharma pipelines—primarily due to the well cited challenges in target validation, valid translational models, and clinical trial design. Unfortunately, this decline in research and development interest has occurred in tandem with an increase in the medical need—in part driven by the success in treating other chronic diseases, which then results in a greater overall longevity along with a higher prevalence of diseases associated with ageing. The lead modality for drug agents targeting the brain remains the traditionally small molecule, despite potential in gene-based therapies and antibodies, particularly in the hugely anticipated anti-amyloid field, clearly driven by the additional challenge of effective distribution to the relevant brain compartments. However, in recognition of the growing disease burden, advanced therapies are being developed in tandem with improved delivery options. Hence, methodologies which were initially restricted to systemic indications are now being actively explored for a range of CNS diseases—an important class of which include the protein degradation technologies.

## 1. Proteolysis-Targeting Chimeras (PROTACs)

Research on central nervous system diseases has decreased due to challenges in target validation and clinical trials, despite a growing medical need [1]. Small molecule drugs still dominate brain-targeted therapies, but interest in gene-based and antibody treatments is rising to address the disease burden [2,3]. Furthermore, advanced therapies and improved delivery options are being explored, including protein degradation technologies [3].

Hijacking the ubiquitin proteasome system (UPS) has been seen as a potential tool for therapeutic use for nearly twenty years [4]. The primary technology used for this targeted protein degradation approach is the utilisation of proteolysis-targeting chimeras, or PROTACs, which are heterobifunctional molecules that are capable of bringing the protein of interest (POI) into proximity with the UPS mechanism. PROTACs are composed of three domains, namely a POI-binding moiety, an E3 ligase-binding moiety, and a molecular linker connecting the two (Figure 1).

### 1.1. Necessary Steps for Targeted Protein Degradation

A specific set of events must take place correctly in order to mimic or harness this process, the failure of any one of which will prevent the targeted degradation of the targeted protein (Figure 2) [5]:

Cellular uptake of the PROTAC to the appropriate intracellular compartment containing the ubiquitination machinery and the POI;Ternary complex formation to enable ubiquitin transfer once the PROTAC is inside the cell. This relies on simultaneously binding to the POI and E3 ligase, which in turn is reliant on the binding affinity of the PROTAC to both proteins. Albeit well described, the thermodynamics of ternary complex formation are less intuitively simple compared to a two-body system. Specifically, a “hook effect”, characterised by a bell-shaped dose–response curve, which occurs when the PROTAC forms 1:1 ligand-bound complexes with either the POI or the E3 ligase, leads to ineffectual complex formation at higher concentrations of the PROTAC, and may give rise to possible issues around in vivo dose selection. Secondary protein–protein interactions (PPIs) may also favour or hinder ternary complex formation through cooperativity or steric clashes, respectively. Figure 3 graphically depicts the concentration-dependent “hook effect”;Once formed, the ternary complex must accommodate the two bound proteins to occupy a favourable conformation, such that ubiquitin transfer may take place to a suitable acceptor site, commonly a surface lysine. Ubiquitin transfer must occur quickly, at a rate faster than dissociation of the ternary complex;Targeted induced polyubiquitination should also kinetically outcompete deubiquitinases, which belong to a large family encompassing wide-ranging substrate specificities;Furthermore, the motif of the transferred ubiquitin residues should facilitate facile recognition through the proteasome to bring about actual degradation;Even if all the previous steps are successful, and the POI is degraded by the proteasome, this does not guarantee a decreased steady state level of protein. The de novo resynthesis rate of the POI, which may vary significantly between cell types, must be markedly slower than the rate of induced degradation. Likewise, the initial reduction of equilibrium protein levels may not persist over time if the loss of mature protein triggers the induction of feedback mechanisms that upregulate either the translation or transcription of the new protein.

#### The Hook Effect

PROTACs sometimes display atypical bell-curve type dose–response curves, which means that they must be evaluated over a range of concentrations. At low concentrations of PROTAC, only a small proportion of POI- and E3 ligase-binding sites are occupied, indicating that only a small number of ternary complexes can form, leading to only some degradation of the POI. At high concentrations of PROTAC, most binding sites of both the POI and E3 ligases are occupied by the PROTAC, again leading to only a small amount of ternary complex formation, and consequently to only some degradation of the POI. An optimum concentration of PROTAC represents a 50% occupancy of all binding sites at 1:1 for POI:E3 ligase, leading to the greatest amount of ternary complex formation, and thus the largest degree of degradation of the POI.

Small-molecule inhibitors act through an occupancy-driven pharmacological model where direct inhibition stops the protein’s function. In contrast, PROTACs work via an event-driven pharmacology model where removal controls protein abundance, indicating that protein function is governed by decreasing the cellular protein level [6].

A key feature of PROTACs that is considered advantageous with respect to small-molecule inhibitors is an added layer of specificity due to the ternary complex formation and their ubiquitin transfer-dependent mechanism of action (MoA). Another advantage is their increased potency compared to a PROTAC’s component inhibitor, along with a sub-stoichiometric drug requirement owing to the catalytic MoA and PPI during ternary complex formation. Furthermore, the degradation of the POI results in prolonged pharmacodynamic suppression in vivo due to protein re-synthesis being required to restore target activity [7]. A range of PROTACs have demonstrated their efficacy at nM concentrations, achieving a high degradation potency due to their MoA [8].

Many proteins, including those which possess non-catalytic functions, namely pseudokinases, transcription factors, non-enzymatic proteins, and scaffolding proteins, lack an active site suitable for an inhibitor and are thus deemed “undruggable”. Moreover, in cancer, inhibition may be less effective or ineffective for small molecules due to the presence of mutations close to the inhibitor binding pocket. PROTACs, however, can allosterically instigate ubiquitination of the POI, circumventing this phenomenon due to their ability to form transient interactions with their targets if ubiquitination occurs quickly within the timescale of ternary complex formation [9].

### 1.2. Blood–Brain Barrier Permeability

Access to the CNS necessitates permeation across the blood–brain barrier. This is a challenge for traditional small-molecule drug discovery, requiring careful consideration of the physiochemical properties e.g., through the use of the MultiParameter Optimisation (MPO) guidelines [10]. This challenge is further amplified for PROTACs due to their increased molecular weight and may be a major limiting factor for their use in the CNS. Radical solutions, such as developing antibody conjugates, have been proposed [11]. However, there are isolated reports of brain penetrable PROTACs. Limited blood–brain barrier permeability was observed for a Tau-targeting PROTAC [12]. However, XL01126, a PROTAC which has been discussed in greater length in Section 2.3, has been shown to be present in the CNS at meaningful concentrations [13]. This is especially surprising, considering the total polar surface area of XL01126 (194.3 Å^2^) relative to the maximum value suggested for reliably entering the CNS (90 Å^2^) [14]. This example gives hope to the field in that access to the CNS is achievable, but the level of the task must not be understated. Additional examples are required in order to spot linker motifs that may allow a common strategy to promote permeability to the CNS.

### 1.3. Solubility of PROTACs

Solubility is a crucial parameter for drug delivery, especially in the context of the CNS, where poor solubility could hamper drug distribution. The proteins of interest that require degradation are intracellularly localised, and thus PROTAC solubility and cell permeability are paramount in their therapeutic success. PROTACs are, by design, large molecules, and an increased MW is often correlated with a poor solubility and a decreased permeability due to larger molecules having a higher propensity for aggregation or precipitation [1]. The molecular structure of the PROTAC itself can determine the extent of its solubility with the presence of polar functional groups, including the hydroxyl (-OH), amino (-NH_2_), or carboxyl (-COOH) groups, increasing its solubility by facilitating hydrogen bond formation with water molecules. Lipophilicity is another factor that needs considering, where the balance between lipophilicity and hydrophilicity needs to be considered in order to achieve an optimal solubility profile. Presently, the current choice for E3 ligase-binding ligands is limited, as explored in Section 1.4, whereas the POI-binding ligand is preliminarily determined by its binding affinity to the POI. Therefore, to modulate the solubility of the PROTAC, varying the linker length and type may be the best course of action [15]. A study by Jiménez et al. of 21 commercial degraders proposed BRlogD and TPSA as key indicators of PROTAC solubility, where 2.58 and 289 Å^2^ were the irrespective calculated thresholds for an experimental solubility classification [16]. However, it should be noted that the PROTACs which display a poor or moderate solubility may benefit from pharmaceutical excipients and formulations.

### 1.4. Choice of E3 Ligase

Currently, only a handful of the broad family of E3 ligases have been exploited, with the most favoured being von Hippel–Lindau (VHL) and cereblon (CRBN), which are part of the Cullin-RING E3 ubiquitin ligase complexes (CRLs) CRL2^VHL^ and CRL4A^CRBN^, respectively [17]. CRBN is the primary target of immunomodulatory imide drugs, whereas VHL mediates the degradation of its substrate HIF-1α [18,19]. Several specialised E3 ligases for potential PROTAC applications have been reported to have an increased tissue specificity, which could be harnessed by the targeted protein degradation modalities to enable the tissue- and cell-type-specific targeting of the disease-causing proteins within the CNS [4,20,21,22]. Among many others, these include the TRIM9 and RNF182 E3 ligases, which are part of the TRIM and RNF families of the E3 ligases, respectively.

Whilst the notion of developing a universal ubiquitin ligase binder would at first seem beneficial, presenting the possibility of a “one-size-fits-all” PROTAC with a customisable POI ligand, this approach is considered as impractical.

Localisation of the PROTAC to specific brain regions poses a potential limitation, with the most widely utilised E3 ligases, CRBN and VHL, being indiscriminately expressed throughout the brain. [23] Many specialised E3 ligases have shown an increased tissue specificity with the potential to be used in a tissue- and cell-specific manner by TPD to degrade the proteins that are involved with diseases in the CNS, but only preliminary steps have been made to develop ligands with the potential to exploit them [4,20,21,22].

As of today, only ~1% of roughly 600 E3 ligases have been successfully targeted by small-molecule degraders (CRBN, VHL, IAPs, MDM2, DCAF15, DCAF16, and RNF114, respectively) [24]. Of the 623 E3 ligases in the human proteasome, around 270 of those are involved with the UPS [25]. PROTACs, with their ability to recruit E3 ligases with tissue-specific expression profiles, have a unique opportunity within a therapeutic context, as in theory they should not degrade the target protein in tissues where the E3 ligase is not expressed. This is particularly important in the context of neurodegenerative diseases because of the clear advantage to only degrading proteins in the brain without causing serious off-target effects in other tissues. Roughly twenty E3 ligases have a narrow expression across human tissues, and of these ligases it is known that four promote the degradation of their substrates (ASB9, KLH10, KLH41, and TRIM69, respectively) [24]. For example, TRIM69 is expressed in the pancreas and testicles, while FBXL16 is solely expressed in the cerebral cortex. Small ligands that bind with a sufficient affinity to one of these E3 ligases can be linked to several ligands that target substrates for degradation within a specific tissue [24].

### 1.5. Role of the Linker

It is known that the PROTAC’s linker plays an important role in protein degradation, the formation of the ternary complex, and the drug’s ADME properties [15,26,27,28]. The first generation of PROTACs used alkyl chains as their linkers [29]. It is vital that drugs possess a sufficient lipophilicity to traverse cell membranes; however, the high lipophilicity of the alkyl chains within the PROTACs reduce their solubility in water. To increase the solubility of the PROTACs, the use of polyethylene glycol (PEG) chains has become commonplace [15,30,31]. This shift from alkyl to PEG chains increases the number of hydrogen-bond donors, thus increasing the solubility. It is incredibly challenging to predict an optimum length of linker—there is no clear correlation between the length, nor the composition, of the PROTAC and its degrading effect [32,33]. Two recent examples of linkers that deviated from the common alkyl and PEG-type chains include ARV-110 and ARV-471, both of which are currently in phase II clinical trials [34,35]. It is interesting to note the increased rigidity obtained by opting for a 1-(piperidin-4-ylmethyl)piperazine linker compared to the first few generations of PROTACs. It could be implied that keeping the number of rotational bonds low with respect to the alkyl and PEG linkers may present an advantage. However, it is wise to be cautious when drawing such a conclusion, as there are only two known examples of this kind of PROTAC linker undergoing clinical trials. It is also possible that other physiochemical factors play a role in the relative success of these PROTACs; for example, increased solubility due to lowering the logD_7.4_ and the introduction of protonatable basic centres.

## 2. Treating Neurodegenerative Diseases Using PROTACs

Recently, several studies have conducted research in exploring the potential use of PROTACs as therapeutic tools for neurodegenerative diseases, namely for Alzheimer’s Disease (AD), Parkinson’s Disease (PD), frontotemporal dementia (FTD), amyotrophic lateral sclerosis (ALS), and Huntington’s Disease (HD). A selection of the most promising studies has been discussed in the following sections.

### 2.1. Tau and Alzheimer’s Disease (AD)

The Tau protein plays a central role in neuronal cells to maintain their cell shape, stabilise microtubules, and provide routes for the transport of cargo proteins. An imbalance in the level of the Tau protein is a principal common factor for a range of neurodegenerative diseases, for example FTD and AD, and mediates the toxicity of amyloid-β (Aβ). The pathology of these diseases has been heavily associated with the toxic accumulation of aberrant Tau species aggregating into paired helical filaments and neurofibrillary tangles, leading to neuronal cell death. Currently, there are no small molecules which are able to modulate its dysregulation as Tau is a non-enzymatic (lacking an active site/pocket) protein [36]. Hence, Tau downregulation is a desired therapeutic strategy [4].

Peptide PROTACs have recently been developed to target the Tau protein for degradation, however, these have been found to suffer from the usual limitations of linear peptides targeting the CNS indications. TH006—a PROTAC peptide (Figure 4)—was revealed to induce Tau degradation in Tau-EGFP-overexpressed N2a cells with Western blotting analysis, albeit at high concentrations (200 µM). Following this, TH006 was used in a mouse model of AD (15 mg/kg intranasal and IV), which revealed reduced levels of Tau in the cerebral cortex and hippocampus [37]. However, the combined use of intranasal and intravenous administration for 10 days to take advantage of olfactory transfer presumable is required due to poor blood-brain barrier (BBB) permeability, which is a common limitation for such peptides. [37,38]. Notably, this work demonstrated a method to regulate, and ultimately degrade, the Tau protein on a post-translational level, which in turn offers a cellular quality-control system to rescue the neurotoxicity of Aβ.

An approach which took inspiration from the abovementioned study demonstrated that a Keap1-dependent peptide PROTAC enables Tau knockdown via the UPS (Figure 5) [39]. A tetramethylrhodamine (TAMRA) analogue was shown to be cell permeable through measurements of fluorescence intensity, taking 12 h to reach its maximal cellular concentration. 

A series of new small-molecule PROTACs have also been developed for targeted Tau degradation. For example, QC-01-175, a Tau-degrading PROTAC based on the PET tracer T807, has been used extensively in vivo to visualise various tauopathies (Figure 6) [40,41]. QC-01-175 was shown to degrade Tau at concentrations ranging from 100 nM to 1 µM in patient-derived neuronal cell models, including the Tau-A152T variant from a PSP patient and the Tau-P301L variant obtained from a behavioural-variant FTD patient [42]. Crucially, such levels of degradation using QC-01-175 were not observed for the normal Tau and low levels of P-Tau observed in wild-type cells, as demonstrated in three iPSC-derived neuronal cell models. 

A follow-up study proposed improvements to the lead PROTAC culminating in two promising Tau-targeting PROTACs with improved activities, namely FMF-06-038 and FMF-06-049, respectively (Figure 7) [43]. The complexities of evaluating these Tau-degrading PROTACs was highlighted, with these PROTACs performing variably across Tau-A152T, Tau-P301L, and WT Tau. Second-generation PROTACs displayed a consistently poor permeability in a CRBN competitive engagement assay; with permeability presenting a major challenge that is more broadly for PROTAC optimisation. Encouragingly, reduced levels of phospho-Tau were observed for up to 8 days following the cessation of treatment, showing that the regeneration of phosphor-Tau is a slow process. Potentially poor brain penetration, lack of clinical biomarkers, and evaluation of off-target binding were all explicitly mentioned as challenges to onward development.

In 2021, Wang et al. designed a PROTAC that was demonstrated to degrade the Tau protein both in vitro and in vivo, preferentially removing pathological Tau proteins via the proteasomal pathway (Figure 8) [44]. C004019 exhibited an exceptionally low cell toxicity in both the HEK293 and SH-SY5Y cell lines, with an DC_50_ of 7.9 nM in a HEK293-hTau overexpressed cell line. Intracerebroventricular administration promoted the clearance of pathological Tau in vivo, and subcutaneous administrations at 3 mg/kg demonstrated a robust and sustained Tau clearance. Most impressively, improved synaptic and cognitive functions were observed in AD-like hTau 3× Tg transgenic mice models. This is particularly notable, as the brain concentration of C004019 was low following its subcutaneous administration (10.8 ng/mL, which translates to a Kpuu value of only 0.009). This illustrates that despite the inherent challenges in achieving a good BBB permeability with PROTACs, meaningful therapeutic interventions can be observed. However, it is also important to note, that administration by intragastrical gavage (20 mg/kg) did not lead to a reduction in Tau, indicating that significant optimisation is needed to achieve an appropriate exposure following oral administration.

Several patents have also been published concerning the treatment of AD using PROTAC technology to target the Tau protein [12,42,45,46], all of which have shown promise so far. However, replication of the results from other laboratories is needed, and further exploration of their viability as a therapeutic treatment for AD is required.

A press release by Arvinas in 2019 claimed that one of their PROTACs in a preclinical model was able to remove 95% of pathological Tau and successfully cross the BBB, crucially without altering the wild-type Tau in the mouse brain 24 h after parenteral administration [47]. However, neither the structure of the small molecule PROTACs nor their corresponding experimental data have been disclosed.

### 2.2. Huntingtin and HD

Huntington’s disease is an autosomal dominant neurodegenerative disorder that is caused by an excessive expansion of a CAG trinucleotide repeat, leading to the formation of polyglutamine-expanded mutant huntingtin (mHTT) protein aggregates which accumulate and eventually lead to cell death. Although multiple different strategies, of varying degrees of success, to develop effective therapies for HD are currently under investigation, a recent small-molecule PROTAC strategy developed by Tomoshige et al. is currently of interest (Figure 9, Tomoshige 1) [48]. Two PROTACs recruiting E3 ligases from the IAP family were shown to induce the ubiquitination and subsequent degradation of mHTT in fibroblast cells derived from two individuals with HD. Further validation with in vivo studies is necessary to obtain a deeper understanding of the full potential for this approach in treating HD, primarily due to wild-type Htt also being degraded. This could suggest that wild-type Htt also forms small oligomers that can be recognised by aggregate binders, leading to PROTAC-mediated degradation [49]. A follow-up study looked at switching to a more potent E3 ligand; however, this did not lead to more potent PROTACs (Figure 9, Tomoshige 7) [50]. This further highlights the complexity of PROTAC design—it is not the binding affinity for the E3 ligase nor the protein of interest that is important, but is rather the stability of the ternary complex, as depicted previously in Figure 3B [51,52].

### 2.3. LRRK2

Mutations within leucine-rich repeat kinase 2 (LRRK2) encoded by PARK8—an implicated gene for PD—have been associated with idiopathic late-onset PD. To date, numerous missense mutations have been linked to PD pathogenesis, including R1441C, R1441G, R1441H, Y1699C, G2019S, and I2020T, and as such LRRK2 has been a popular choice of target for therapeutic modulation [53,54]. Recently, a study by Konstantinidou et al. disclosed that work had commenced on designing and synthesising a PROTAC capable of degrading LRRK2 [55]. Although cell permeability and target binding were observed, these PROTACs were unable to induce ubiquitination and subsequent degradation, with difficulties in assembling the ternary complex having been postulated as a possible reason. A follow-up patent was later filed claiming selective modulators of mutant LRRK2 proteolysis, suggesting that progress has been made regarding the CRBN-based G2019S-LRRK2-PROTAC degraders (wit G2019S-LRRK2 being the most common LRRK2 pathogenic mutation) [56].

Recently, Ciulli et al. published a substantial piece of work on LRRK2 PROTAC degraders, culminating in XL01126 (Figure 10) [13]. Not only is the lead compound a potent degrader of LRRK2, but it also exhibited favourable properties in both in vitro and in vivo experiments (C57BL/6 mice). Despite having a poor solubility, XL01126 has an oral bioavailability of 15% in rats with a half-life of 21.9 h—most likely protected from metabolism by its high degree of plasma protein binding. Moreover, XL01126 was detected at a concentration of 14 nM in both the brain and the CSF. This ability to cross the BBB is perhaps unexpected considering its high total polar surface area (194 Å^2^). 

### 2.4. Alpha-Synuclein

Another opportunity to target a range of neurological diseases where PROTAC technology may offer improvements over current therapeutic methods is in targeting α-synuclein. α-Synuclein is a protein that accumulates within the neurons of individuals with PD, leading to the formation of Lewy bodies [57]. It is an intrinsically disordered protein that drives the pathology of PD, and for which targeted protein degradation via the proteasome may offer a therapeutic route.

A synthetic peptide PROTAC has recently shown a selective degradation of α-synuclein via the proteasome in a time- and dose-dependent manner within neuroblastoma cells and primary neurons [58]. This study demonstrated that, functionally, the reduction in α-synuclein rescued mitochondrial dysfunction and cellular defects, which were consequential of α-synuclein overexpression, indicating at the possibility of utilising the peptide PROTAC as a potential strategy to treat PD. Further validation is required to determine whether this method has potential clinical applications.

In 2020, Kargbo summarised a series of small-molecule PROTACs (Figure 11) developed by Arvinas that are capable of degrading up to 65% of α-synuclein at a concentration of 1 µM in various cell lines [59,60]. These small-molecule PROTACs consisted of an α-synuclein-binding domain, a linker, and different E3 ligase-binding domains, namely for the VHL, CRBN, IAP, and MDM2 E3 ligases. Independent reproduction of the results is required in order to fully realise the claims made therein; however, it is an encouraging starting point in exploring a targeted protein degradation therapeutic approach using small-molecule PROTACs in tackling PD.

### 2.5. C-TDP-43

The TAR DNA-binding protein 43 (TDP-43) is another example of a misfolded protein implicated in a number of neurodegenerative diseases, such as amyotrophic lateral sclerosis (ALS), frontotemporal dementia (FTD), and limbic-predominant age-associated TDP-43 encephalopathy (LATE) [61]. The successful removal of the toxic C-terminal form by the use of a PROTAC has recently been reported [62]. JMF4560 (Figure 12) displayed selectivity over the endogenous TDP-43, with Western blot analysis showing a near-complete depletion at 5 µM in a cellular model.

As with non-brain diseases, PROTACs are the leading technology that is currently being exploited, with other modalities still at significantly earlier stages. This pattern is replicated in the field of CNS diseases, with wider non-PROTAC technologies only recently starting to be reported.

## 3. Antibody PROTACs

PROTACs have shown great promise in the treatment of a range of CNS diseases. However, they are not generally tissue specific due to the inclusion of E3 ligases with expansive expression. Despite the discovery of numerous ligases with an attested tissue specificity, including brain specific ligases, none of these ligases have yet been utilised to develop an effective tissue-specific PROTAC [63,64,65,66]. The development of a tissue- or cell-specific PROTAC could prevent protein degradation in normal cells and significantly reduce the side effects of these compounds.

The inclusion of antibodies in PROTAC technology is a new and emerging field of research, which has proven to be effective in providing these compounds with an enhanced tissue and cell specificity. There are two major categories of antibody PROTACs—antibody–PROTAC conjugates and antibody-based PROTACs (AbTACs) [67]. 

Antibody–PROTAC conjugates were derived from the existing antibody–drug conjugates but have the added benefits of the small-molecule PROTACs [68]. This new class of drugs consists of PROTACs connected to cell-specific antibodies through a cleavable linker and directs the PROTACs to specific cells for the internalisation and degradation of the POI. Maneiro and colleagues demonstrated the specificity of these drugs by creating an antibody–PROTAC conjugate which degraded bromodomain-containing protein 4 (BRD4) specifically in HER2-positive breast cancer cell lines without affecting the BRD4 levels in HER2-negative cells. Their antibody-dependent specificity was further confirmed by assessing the same PROTAC molecule with the antibody removed. This PROTAC demonstrated an effective degradation in both cell lines, thereby confirming the antibody’s ability to provide cell specificity [69]. Dragovich and others also developed several antibody–PROTAC conjugates targeting ERα in HER2-positive cells and gave evidence for efficient intracellular degrader release in the HER2 cells alone [70]. In addition to tissue selectivity, there a number of other potential advantages to these compounds, such as improved pharmacokinetic properties, and simpler routes for administration [71,72].

AbTACs are recombinant bispecific antibodies and differ from antibody–PROTAC conjugates since they possess the ability to bind to both the POI and an E3 ubiquitin ligase without the need for additional components. These antibody-based PROTACs use the lysosomal pathway for the degradation of their targets and could allow a wider range of applications due to their ability in being able to degrade challenging membrane proteins [73]. Cotton and colleagues recently reported the first AbTAC, which allowed the degradation of the cell-surface protein PD-L1 (programmed death-ligand 1). This study presented recombinant AbTACs with the ability to degrade 63% of the protein in vitro through its colocalisation with the RNF43-membrane-bound E3 ligase [73,74]. Despite the capability of both types of antibody compounds, there are currently no reports of AbTACs targeting the central nervous system. This may be due to the difficulty of developing cell-specific antibodies which can penetrate the blood–brain barrier along with the high manufacturing costs of these constructs [71]. However, there have been a number of brain penetrant antibodies showing great promise in clinical trials, which could aid the development of specific and brain penetrant antibody PROTACs [75].

## 4. The Potential of New Degrader Technology

Despite the potential ability of PROTAC technology in being able to target many of the proteins mentioned above, there are many limitations to their use as pharmaceutical compounds, particularly with regard to targeting CNS diseases. PROTACs are classically large compounds with high molecular weights (>800 kDa) and high polar surface areas, which gives rise to particular limitations, including their low solubility, poor cell permeability, low oral bioavailability, and critically, poor penetration of the BBB [76]. Another clear limitation with PROTACs is their inherent dependence on the choice of E3 ligase subunit, which evidently can limit their application in particular cell types. BBB permeability and efficiency may be impacted by varying molecular weights, modulated by a range of linker lengths and types, different POI-binding ligands, and differing E3 ligase-binding ligands. Localisation of the PROTAC to specific brain regions could also be a potential limitation, with the most widely utilised E3 ligases, CRBN and VHL, being indiscriminately expressed throughout the brain [23]. The dependence on E3 ligase subunits can also lead to cancer cell resistance following chronic PROTAC treatment, as demonstrated in a study by Zhang et al., where cancer cells acquired resistance to both VHL-based and CRBN-based BET PROTACs [77,78]. Lastly, as the PROTAC technology depends on degradation via the intracellular UPS system, many proteins, including extracellular protein targets, cannot be pursued by PROTACs [79]. Extracellular proteins, which account for approximately 40% of the proteome, and other biomolecules cannot be targeted through accessing the UPS system. These biomolecules come in a variety of significant classes with differing roles, such as growth factors, proteins with expanded repeat sequences, protein aggregates, and other non-protein molecules, which are of high importance in disease progression [77,80,81].

## 5. Hydrophobic Tags

The use of hydrophobic tags (HyTs) presents an alternative approach to PROTACs for the degradation of proteins, but also utilises the proteasomal pathway to conduct its activity. Hydrophobic tag degraders are bifunctional molecules with a ligand for the protein of interest (POI) attached to a hydrophobic tag, such as adamantane or *tert*-butyl carbamate-protected arginine (Boc_3_Arg). Partially denatured proteins, where the core hydrophobic residues are exposed, are quickly degraded by the UPS system [82,83]. The hydrophobic tag, which mimics the exposed hydrophobic core residues of a denatured protein, interacts with the chaperone proteins, heat shock proteins 70 and 90 (HSP70/90), and these direct them to the CHIP E3 ligase for protein ubiquitination and proteasomal degradation [84]. HyTs offer advantages over PROTACs as they usually have smaller molecular weights and fewer hydrogen bond donors and hydrogen bond acceptors, leading to more favourable properties for CNS drugs (in particular with regard to the penetration of the blood–brain barrier) [85]. 

Several hydrophobic tag degraders have been developed for the treatment of CNS diseases. Gao and colleagues synthesised a hydrophobic degrader which targeted the Tau protein associated with Alzheimer’s disease and other tauopathies. This degrader contained adamantane as the hydrophobic tag, YQQYQDATADEQG peptide as the Tau binding ligand, a GSGS peptide linker, and a poly-D-arginine cell-penetrating peptide. A reduction in the level of Tau in vitro and in the brain of AD mouse models was observed by this degrader, as well as an effective cell penetration in wild-type N2a cells and their ability to cross the BBB [86]. 

In addition to this, Gao and colleagues also developed several single and double hydrophobic tag degraders to target the amyotrophic lateral sclerosis (ALS)-associated protein, TDP-43. Among these degraders was the peptide-based degrader D4, which showed the strongest degradation potential in cells and in a transgenic drosophila model, as well as a low cytotoxicity. D4 is composed of two adamantane groups as hydrophobic tags, an EDLIIKGISV peptide TDP-43 recognition motif, a KGSGS peptide linker, and a GRKKKRRWRRR cell-penetrating peptide. Despite the degradation activity achieved in vivo in this study, high doses of 20–150 µM were required for the different hydrophobic tag degraders activity in cells, and several cytotoxicity issues were also observed for some of these degraders [49,87]. Hirai and colleagues have also developed a hydrophobic tag degrader targeting the disease-causing mutant huntingtin protein, from what was originally a PROTAC [85]. The conversion of this PROTAC to a hydrophobic tag degrader decreased the molecular weight and reduced the number of hydrogen bond donors and acceptors, which in turn improved the permeability into the CNS whilst retaining the drug’s potency. This study allowed for the discovery of a potentially brain-penetrant degrader of mHTT through IAM chromatography analysis and an in vivo brain penetration assay. This hydrophobic tag consisted of an adamantane hydrophobic tag and a known mHTT aggregate-binding moiety. Their results proved to be successful with the development of a brain permeable HyT capable of inducing the selective degradation of mHTT of up to 60% at concentrations of 5 µM. Selective degradation, however, could not be achieved at higher concentrations. This study shows promise and highlights the advantages of hydrophobic tag degraders as an alternative approach for targeting CNS diseases. However, as was the case for the initial PROTAC work, further studies are required to demonstrate the potential of this approach, and in particular to replicate the current single-case examples that have been published.

## 6. The Lysosomal Pathway

In order to target the biomolecules that cannot be degraded by the UPS pathway, efforts have been made to create alternative degraders that can access the lysosomal degradation pathway, which include the lysosomal-targeting chimeras (LYTACs), autophagy-targeting chimeras (AUTACs), and autophagosome-tethering compounds (ATTECs). The lysosomal pathway is important in regulating extracellular and intracellular homeostasis in cells, as well as for many other processes. Degradation by the lysosome can occur using two major pathways: autophagy and the endosome–lysosomal pathways [88]. For the endosome–lysosomal pathway, substrates, such as extracellular and membrane-bound proteins, enter the cell by interacting with a recycling receptor triggering endocytosis, a process in which the plasma membrane invaginates to enclose the substrate (bound to the receptor) in a vesicle [89]. Then, early endosomes are fully formed in the cytoplasm, and their slightly acidic pH means that the receptors dissociate from the cargo and are returned by the recycling endosomes to the surface, where they can be reused and allow for the endocytosis of further cargo. Meanwhile, the internalised cargo is transported to the late endosomes and then to the lysosome [90]. The lysosome contains numerous acidic hydrolases that allow for the breakdown of the cargo substrate by hydrolysis. This process is summarised in Figure 13.

The degradation of cargo through the autophagy pathway uses a slightly different mechanism, but also ends with the hydrolysis of cargo by the lysosome. Autophagy is a complex process involving numerous molecules and signalling events. Initially, various triggers can lead to the nucleation of a phagophore, where certain cytoplasmic compounds are enveloped by a double membrane. LC3 receptor-type proteins insert themselves into the phagophore membrane, and the cargo is degraded both non-selectively and selectively. In selective autophagy, specific cargo (K63-ubiquitinated proteins) will interact indirectly with the LC3-II membrane-bound proteins through adaptor proteins, such as p62/SQSTM1, and become enclosed as the phagophore develops [91,92]. In bulk (non-specific) autophagy, random cytoplasm containing non-specific cargo is isolated and fully enclosed as the phagophore elongates and the edges fuse to give an autophagosome vesicle [93]. Following the elongation of the phagophore to give the autophagosome, this structure fuses with the lysosomes to form an autolysosome. The lysosomal acidic hydrolases then degrade the enclosed cargo and the inner membrane; the resulting products are recycled back to the cytosol [94,95,96]. This process is summarised in Figure 14.

## 7. Lysosomal-Based Degraders

Degraders that use the lysosomal pathway are based on a similar concept to PROTACs in the way that they are composed of a ligand that binds to the molecule of interest and a ligand that triggers its degradation, usually tethered together by a linker. There are different types of compounds that have different properties which can be used to degrade the different types of biomolecules, which will be summarised in this review.

### 7.1. LYTACs

Different types of LYTACs (lysosomal-targeting chimeras) have been developed which can target extracellular and membrane-bound proteins [77]. These LYTACs consist of a moiety that binds to the substrate molecule, which is attached to a ligand that can bind to a lysosome-shuttling receptor [76]. The substrate protein bound to these LYTAC compounds can undergo internalisation via clathrin-mediated endocytosis; this is triggered by the LYTAC’s interaction with a membrane-bound receptor and involves the membrane forming around the protein–LYTAC–receptor complex [88]. The substrate then progresses through to the early endosomes and late endosomes, and this then fuses with the lysosome where the substrate is degraded, as previously described. The Bertozzi lab developed the first-generation LYTAC compounds that contained a 20- or 90-mer of mannose-6-phosphate (M6P), a glycan ligand, as the receptor binding ligand. This LYTAC has been stated to bind non-covalently to the substrate and allow its transport into the cell via interactions with the M6P ligands with the cation-independent M6P receptors (CI-M6PR) [97].

This concept was further explored with the development of LYTACs with target-specific antibodies; these different antibodies were conjugated to a poly(M6Pn)-bearing glycoprotein to form target-specific LYTACs. Bertozzi and colleagues further demonstrated the potential of LYTACs as therapeutics, as they showed their ability to degrade the membrane protein epidermal growth factor receptor (EGFR), CD71, and programmed death-ligand 1 (PDL1) [97]. In addition to this, a liver-specific LYTAC was developed by Tang and colleagues that permitted endocytosis and lysosomal degradation by binding to the asialoglycoprotein receptor (ASGPR) [98]. Whilst LYTACs are uniquely able to degrade extracellular proteins, they suffer from major practical considerations related to their significant size, as shown in Figure 15, resulting in a poor tissue permeability and a low CNS penetration. In addition to this, the LYTACs’ receptor binding ligand (M6P) is challenging to synthesise, and their peptide-like structure mean that immune responses were hypothesised to be an issue in vivo [43,68,69]. The antibody-based LYTACs also have some additional challenges, including the high cost, potential immunogenicity, and lack of understanding about the endocytosis mechanism [99,100,101].

### 7.2. AUTACs

Autophagy-targeting chimeras (AUTACs) are bifunctional molecules that degrade their targets by accessing the previously mentioned autophagy pathway. Similar to the PROTAC’s structure, they contain a specific ligand for the target molecule that is attached by a linker to a ligand that triggers ubiquitination. However, instead of triggering the K48 polyubiquitination of the substrate as for the PROTACs, AUTACs trigger K63 polyubiquitination; this type of polyubiquitination leads to the degradation of the substrate through the autophagy pathway. The degradation tag commonly used in these molecules is a guanine derivative; this mimics the S-guanylation of the invading group A streptococci (GAS) by the endogenous nucleotide 8-nitroguanosine 3′,5′-cyclic monophosphate (8-nitro-cGMP). S guanylation is a post-translational modification that marks the invading GAS for K63 polyubiquitination and subsequent degradation [76,102]. Once the substrate has been ubiquitinated it is recognised by p62/SQSTM1, a ubiquitin-binding receptor, and is shuttled to the phagophore. In the autophagosome, the p62/SQSTM1 receptor binds to LC3-II, a ubiquitin-like membrane-bound protein which docks the ubiquitinated substrate into the phagophore [103].

In a study by Takahashi and colleagues, AUTACs were developed which downregulated cytosolic proteins and improved mitochondrial turnover by degrading the small, fragmented mitochondria. The mitochondria-degrading AUTAC was able to significantly improve mitochondrial activities in down syndrome (DS)-derived fibroblast cells [102,104]. Chang and colleagues developed different a AUTAC technology, which allows for the binding to the ZZ domain of the p62/SQSTM1 receptor [105]. This technology allowed for the degradation of various oncoproteins and degradation-resistant aggregates under conditions of neurodegeneration at nanomolar DC_50_ values in vitro and in vivo. One application of this technology was to degrade the aggregation prone P301L Tau mutant to treat Alzheimer’s disease. The AUTACs PBA-1105 and PBA-1106 (Figure 16) were able to induce the degradation of mutant Tau in SH-SY5Y cells at a DC_50_ of approximately 1–10 nM and effectively removed aggregated Tau from mouse brain. In addition to this, these AUTACs (shown in Figure 16) were also able to induce the lysosomal degradation of both the nucleus- and cytosol-resistant mutant huntingtin, suggesting that AUTACs could be applied to target a range of protein aggregates in CNS diseases [105]. Similar types of AUTAC degraders have also been developed to degrade many other proteins, such as MetAP2, FKBP12, BET, and TSPO, demonstrating the potential of this technology in cancer, neurodegenerative diseases, and several other diseases [106]. In spite of the success and many advantages of AUTACS, including its broad and specific spectrum of targets, this technology is in its infancy and independent replications of these key findings are required. Currently, the mechanism of selective degradation is still not fully understood, and it is still to be determined whether AUTACs affect global autophagy or any other cellular functions [101].

### 7.3. Molecular Glues and ATTECs

Molecular glues are an intriguing alternative to PROTACs that have several distinct advantages due to their mechanism of action [107]. Molecular glues do not require a lipophilic pocket in either the POI or the ligase, but rather promote direct protein–protein interactions between both. This can result in simplified chemical structures that adhere more closely to the typical drug discovery physiochemical properties. This has obvious benefits considering their pharmacodynamic properties, but perhaps more importantly in the context of this review, they stand a greater chance of achieving blood–brain barrier permeability due to their lower MW and TPSA. One critical drawback of molecular glues is that their rational design is currently near-impossible as they are often found serendipitously. Indeed, there are instances where they have been discovered by accident in PROTAC research [108,109]. Care is needed so as not to assume biological changes following treatment with PROTACs are always due to their intended mechanism of action, in addition to maximising the potential of molecular glues when they are found. 

Autophagy-tethering compounds (ATTECs) are a specific class of molecular glues that can degrade proteins and other molecules through the autophagy pathway but access the autophagy pathway using a different mechanism to the AUTACs. Unlike AUTACs, the ATTECs’ activities are not dependent on ubiquitination, and they offer a more direct approach for degrading the molecules of interest, including biological molecules other than proteins by autophagy [76]. These molecules exert their role by tethering the molecule of interest directly to the autophagosome via the LC3 membrane protein present on nascent autophagosomes (phagophores) [106]. This technology has been applied to target Huntington’s disease and the mHTT (mutant huntingtin) protein. A few different ‘molecular glues’ like ATTECs were developed by Zhaoyang and colleagues that were able to degrade the mHTT protein and not the wild-type protein due to their interactions with the extended polyglutamine (PolyQ) sequence on the mHTT protein. Two of these compounds were found to be effective for the degradation of mHTT in vivo HD mouse models at nanomolar concentrations and were also able to cross the BBB (10O5 and AN2, Figure 17) [110]. These compounds were investigated further to see whether they could target other pathogenic species with extended polyQ sequences, and 10O5 and AN2 were both found to be effective in degrading the mutant ataxin-3 (ATXN3) protein in fibroblasts derived from patients with the neurodegenerative disease spinocerebellar ataxia type 3 [111]. In addition to this, these compounds have been applied to degrade lipid droplets, which function as lipid-storing locations in the cell. Yuhua and colleagues developed bifunctional ATTEC molecules consisting of the LC3-binding motifs (used to degrade mHTT) and a lipid droplet-binding motif, derived from oil red O, which were connected by linkers [112]. These compounds were able to degrade lipid droplets in cells, which was not possible with the use of the PROTACs or AUTACs. This demonstrates the potential of this technology to degrade non-protein molecules [113,114]. These molecules offer a large advantage over other technologies when targeting CNS diseases, due to their low molecular weights and drug-like properties, meaning that they are more likely to penetrate the BBB. The ability to target a wider range of molecules, including those other than proteins, is additionally attractive. However, there have only been a few reports published on this topic, and further examples are needed to gain confidence in the ultimate utility of this technology. Furthermore, additional LC3-bound chemical moieties need to be discovered, and further studies need to be conducted on designed chimeras for the advancement of this technology [76].

### 7.4. Chaperone-Mediated Autophagy Degraders

Chaperone-mediated autophagy degraders have also been developed to target CNS diseases [84]. These degraders contain a specific protein-binding ligand and a chaperone-binding ligand, such as the pentapeptide KFERQ motif and the VKKDQ peptide found on α-synuclein. The chaperone-binding motif present on these degraders bind to the heat shock cognate 71 KDa (HSC70) chaperone protein and other co-chaperones, which allows for the delivery of the degrader and the associated protein to the lysosome for degradation [115]. A few examples of chaperone-mediated autophagy degraders have been developed, with one example by Fan and colleagues targeting α-synuclein [116,117]. This degrader contained a cell-penetrating peptide (CPP) sequence TAT, a short amino acid sequence binding α-synuclein (βsyn36), and the chaperone-mediated autophagy targeting motif (CTM) KFERQ sequence. This chaperone-mediated autophagy degrader demonstrated an effective degradation of the protein in rat neuronal cultures and in rats’ brains in vivo. Chaperone-mediated autophagy degraders have also been employed by Bauer and colleagues to degrade mHTT in the mouse brain. Expression of a 46 amino acid peptide adaptor molecule compromising two copies of the polyglutamine-binding peptide 1 (QBP1) sequence and the two KFERQ and VKKDQ HSC70-binding motifs were used to achieve this degradation and improve motor impairment [49]. This technology has potential to be adapted to therapeutics for various CNS diseases but is still at its earliest stages.

As with all new technologies, the ultimate successful application requires navigation of the well-described hype-hope cycles, and this is particularly true in areas of disease where there are no chances of disease-modifying therapies in the near- to mid-term. The ability to remove proteins by the use of PROTAC technologies has clearly opened enormous opportunities within various cancers—with a number of programmes having been evaluated in patient-based studies. The extension of protein degradation to aggregate removal is of particular interest in the neurodegenerative field, and the small-molecule approaches here may well offer distinct advantages over the current antibody approaches. However, the additional challenges involved in successfully progressing these PROTAC oncology projects to the clinic should not be overlooked—and a clear appreciation is required of the uphill struggle that is needed to be able to bring a CNS-targeted PROTAC to the same development stage. However, the growing efforts in this area are encouraging, and hopefully through this collective effort, breakthroughs will be made that will allow for the therapeutic potential of these approaches to be realised.

## Figures and Tables

**Figure 1 biomolecules-13-01164-f001:**
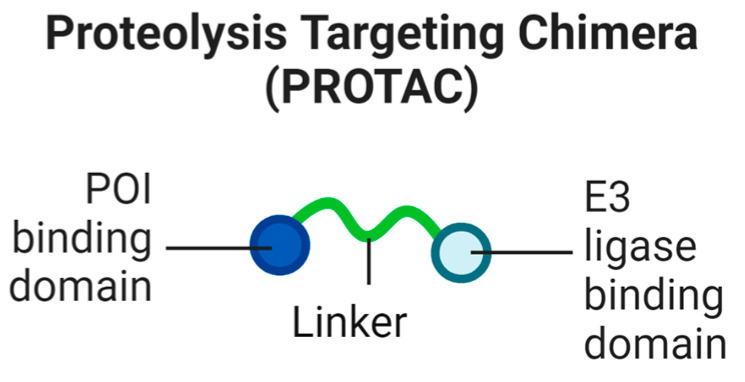
The constituent parts of a PROTAC.

**Figure 2 biomolecules-13-01164-f002:**
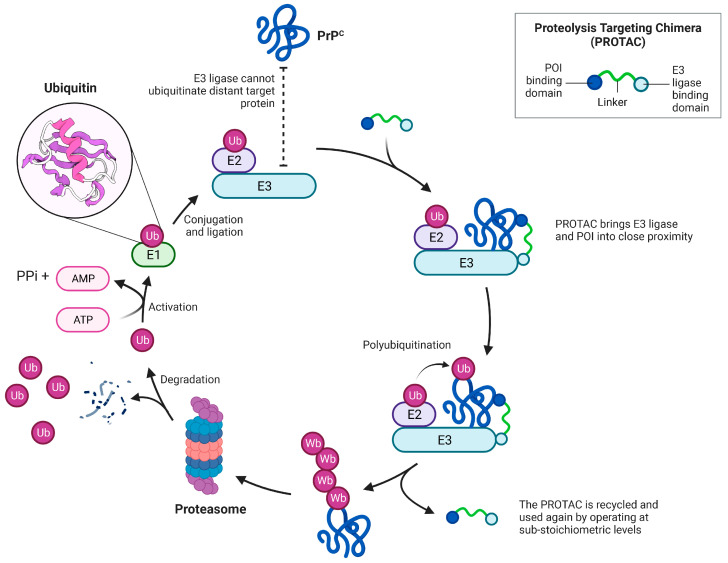
Schematic of a PROTAC bringing the POI into proximity with the UPS.

**Figure 3 biomolecules-13-01164-f003:**
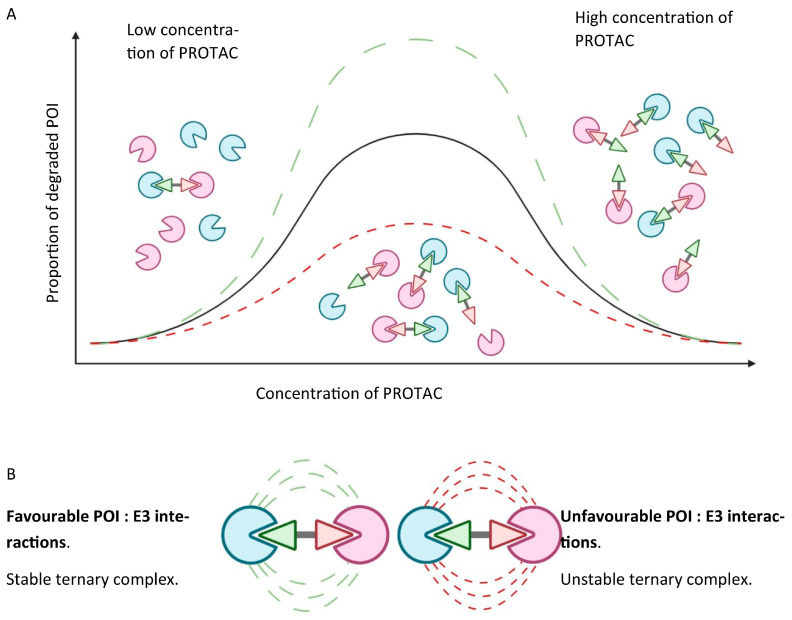
(**A**) Hook effect depicting the proportion of degraded POI when the systemic concentration of PROTAC is both too high and too low. (**B**) PPIs between the POI and E3 ligase. Importantly, even when PROTAC–POI affinity is weak, favourable PPIs can stabilise the ternary complex, leading to effective ubiquitination and consequent degradation.

**Figure 4 biomolecules-13-01164-f004:**
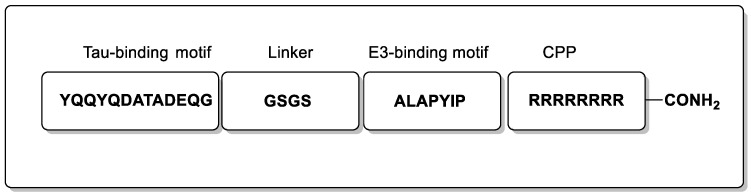
Representation of the peptide PROTAC TH006.

**Figure 5 biomolecules-13-01164-f005:**
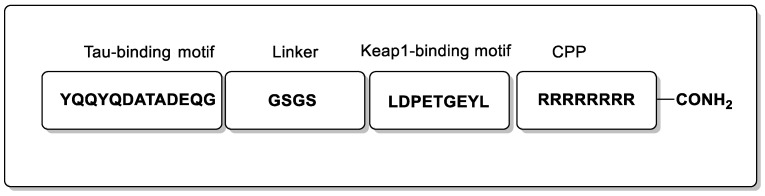
Representation of a Keap1 peptide PROTAC.

**Figure 6 biomolecules-13-01164-f006:**
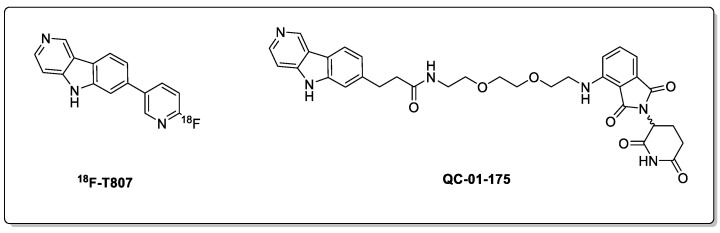
Tau PET tracer and PROTAC analogue.

**Figure 7 biomolecules-13-01164-f007:**
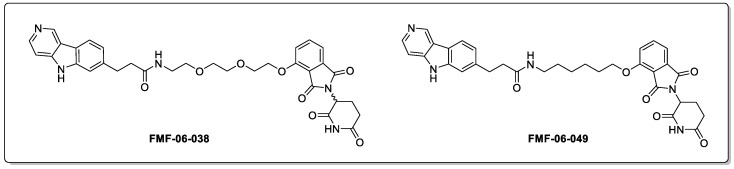
Second-generation Tau PROTAC degraders.

**Figure 8 biomolecules-13-01164-f008:**
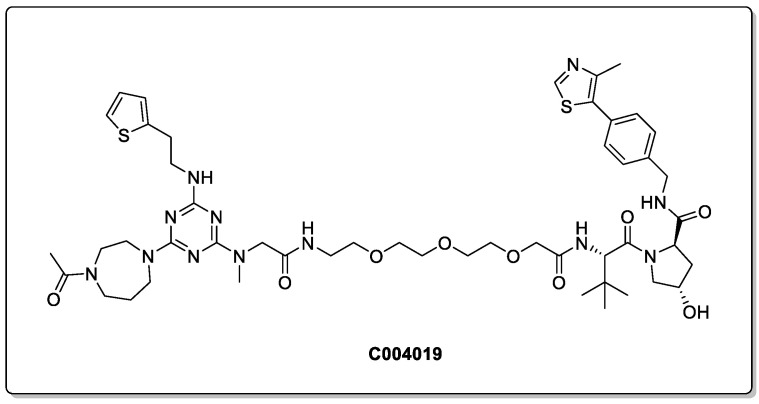
Effective in vitro and in vivo Tau-degrading PROTAC.

**Figure 9 biomolecules-13-01164-f009:**
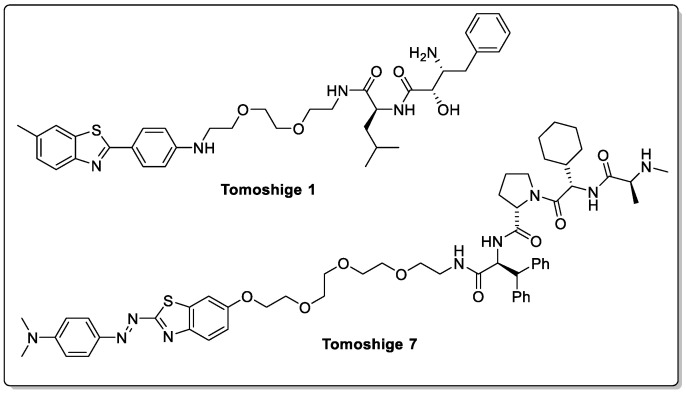
Two generations of mHTT-degrading PROTACs by Tomoshige et al., which showed an initial promise.

**Figure 10 biomolecules-13-01164-f010:**
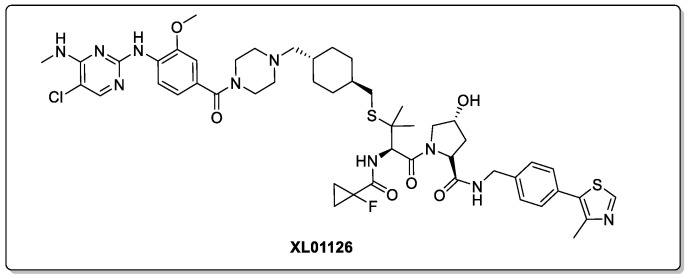
PROTAC targeting LRRK2 for degradation.

**Figure 11 biomolecules-13-01164-f011:**
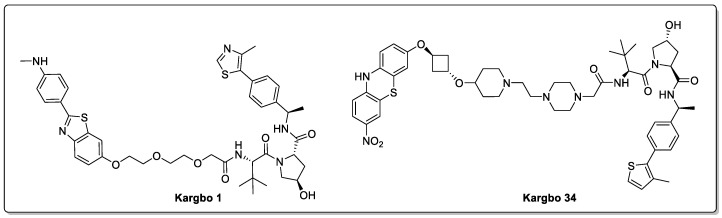
α-synuclein-degrading PROTACs.

**Figure 12 biomolecules-13-01164-f012:**
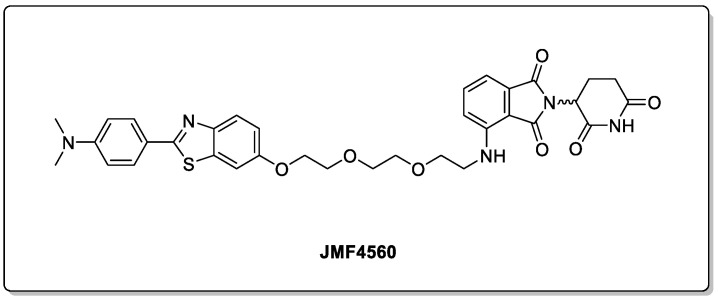
C-TDP-43-degrading PROTAC, JMF4560.

**Figure 13 biomolecules-13-01164-f013:**
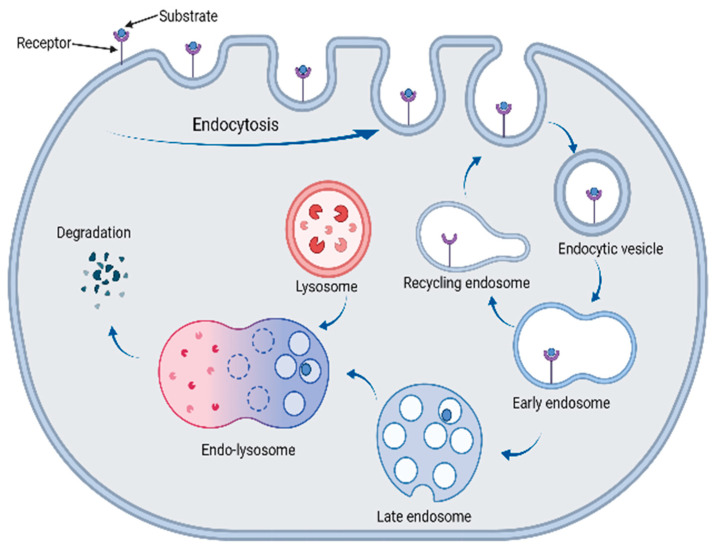
The degradation of a substrate via the endosome–lysosomal pathway.

**Figure 14 biomolecules-13-01164-f014:**
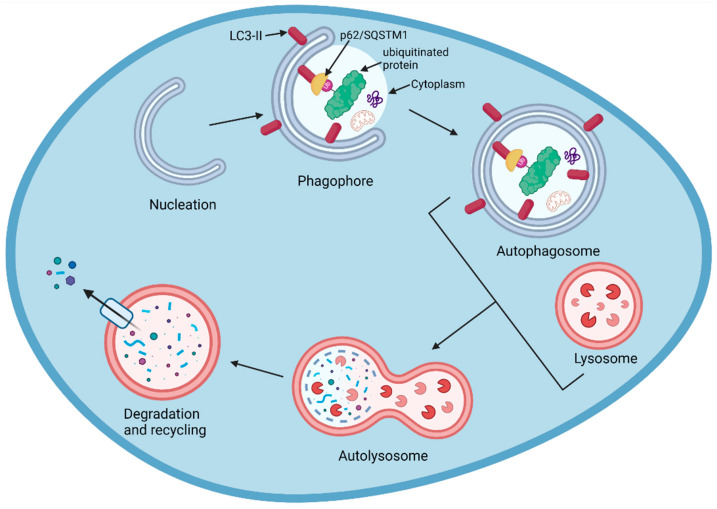
The non-specific bulk and specific degradation of cargo via the autophagy pathway.

**Figure 15 biomolecules-13-01164-f015:**
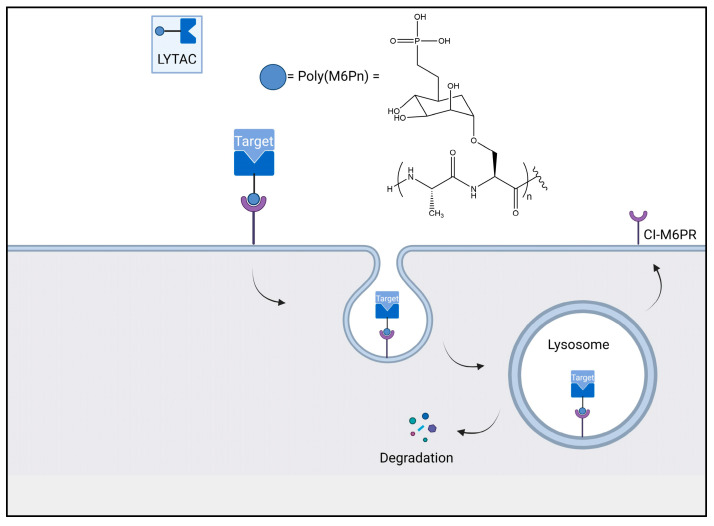
The degradation of substrates using LYTACs.

**Figure 16 biomolecules-13-01164-f016:**
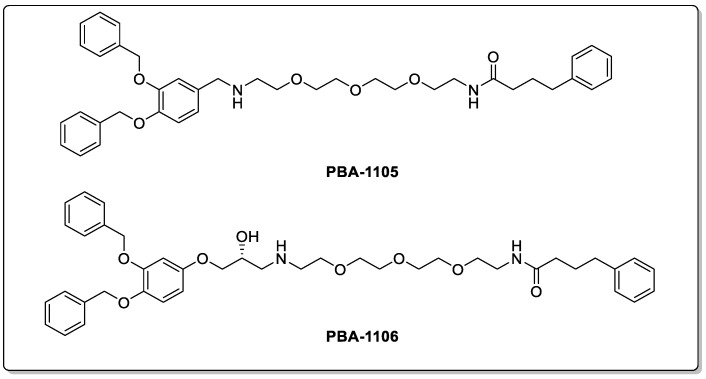
AUTACs targeting mutant Tau for degradation.

**Figure 17 biomolecules-13-01164-f017:**
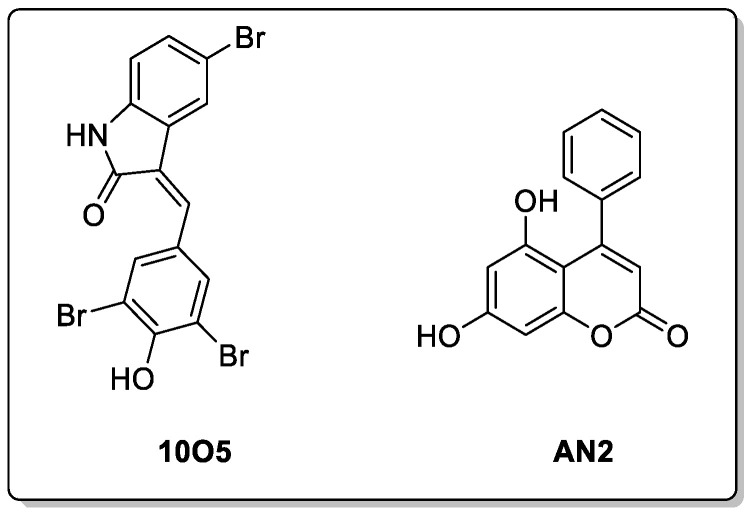
Effective ATTECs targeting mHTT shown to cross the BBB.

## Data Availability

Not applicable.

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
