# Peer review of "Central Nervous System Targeted Protein Degraders"

_biomolecules, 2023, doi:10.3390/biom13081164_

Round 1
Reviewer 1 Report
Despite the potential of gene therapies and antibodies, small molecules remain the gold standard for CNS-targeting drugs, owing to the inherent challenge of effectively delivering alternative agents to the brain. However, advancements in delivery methods are being concurrently pursued with the development of novel therapies, including protein degradation technologies such as PROTACs. PROTACs have demonstrated promise in cancer treatment and have garnered substantial interest in the context of neurodegenerative diseases. Nonetheless, the journey toward clinical application is fraught with challenges, particularly for CNS-targeted PROTACs. In this review manuscript, authors present collective endeavors in the design, mechanism of action, and therapeutic potential of this advanced technology by expanding the knowledge borders and embracing also the details of LYTACs, AUTACs, and ATTECs.
The review topic is intriguing, and I appreciate the effort you have put into this work. However, there are several points that I believe need to be clarified or expanded upon to improve the quality and impact of the manuscript.
1- Blood-brain barrier permeability: Could you provide more detailed information on how PROTACs can traverse the blood-brain barrier? This is a critical point for any therapeutic strategy targeting the CNS, and further clarity on this matter will be very useful for your readers.
2- Length, molecular composition, and potential cross-reactivity of linkers: The manuscript would greatly benefit from a more detailed explanation about the linker's characteristics. It would be particularly useful to understand the factors influencing the choice of length and molecular composition and how these characteristics may affect the PROTAC's function and specificity.
3- Unique and common features between molecular glues and PROTACs: A comparative discussion of these two technologies will help to contextualize the potential advantages and disadvantages of PROTACs in CNS diseases. Could you discuss the features that make PROTACs distinct and those they share with molecular glues?
4- Antibody-PROTACs: Please elaborate on the role of antibody-PROTACs in your proposed therapeutic strategy. The reader would appreciate a more in-depth discussion on how these constructs work and what their potential advantages and disadvantages might be.
5- Solubility of PROTACs: Please provide more information about the solubility of PROTACs. Solubility is a crucial parameter for drug delivery, especially in the context of the CNS, where poor solubility could hamper drug distribution.
6- Potential to use a universal ubiquitin ligase binder: It would be interesting to see your perspective on the possibility of using a universal ubiquitin ligase binder, including the potential advantages, disadvantages, and current evidence supporting this approach.
Overall, I believe that the study represents a valuable contribution to the field. However, addressing these issues will undoubtedly strengthen the manuscript. I look forward to your responses and the revised manuscript.
The English language is fine. Minor editing and grammatical corrections are suggested.
Author Response
1- Blood-brain barrier permeability: Could you provide more detailed information on how PROTACs can traverse the blood-brain barrier? This is a critical point for any therapeutic strategy targeting the CNS, and further clarity on this matter will be very useful for your readers. A short section on blood brain barrier permeability has been added (1.2, line 111-125). There is not a great deal of data on the ability of PROTACs to traverse the blood brain barrier, but two examples of brain-penetrant PROTACs are included, along with potential of using antibody conjugates. There is not sufficient data at the moment to suggest a common strategy e.g. a type of linker that would improve the chances of bbb permeability.
2- Length, molecular composition, and potential cross-reactivity of linkers: The manuscript would greatly benefit from a more detailed explanation about the linker's characteristics. It would be particularly useful to understand the factors influencing the choice of length and molecular composition and how these characteristics may affect the PROTAC's function and specificity. A brief section has been added to address this, charting the move away from initial very lipophilic linkers to polyethylene glycol chains and finally the piperazine linkers present in more advanced PROTACs (1.5, line 182-202).
3- Unique and common features between molecular glues and PROTACs: A comparative discussion of these two technologies will help to contextualize the potential advantages and disadvantages of PROTACs in CNS diseases. Could you discuss the features that make PROTACs distinct and those they share with molecular glues? The concept of molecular glues are included (lines 600-612) before moving on to discussing ATTECs. Their advantages and disadvantages relative to PROTACs are discussed in addition to reminding the reader that it is possible that some designed PROTACs in fact act as molecular glues.
4- Antibody-PROTACs: Please elaborate on the role of antibody-PROTACs in your proposed therapeutic strategy. The reader would appreciate a more in-depth discussion on how these constructs work and what their potential advantages and disadvantages might be. An extensive section on antibody-PROTACs has been included (3, lines 371-412).
5- Solubility of PROTACs: Please provide more information about the solubility of PROTACs. Solubility is a crucial parameter for drug delivery, especially in the context of the CNS, where poor solubility could hamper drug distribution. A section on solubility is included (1.3, lines 127-143), which discusses parameters to keep in mind when optimising solubility of PROTACs. It is discussed that the linker might be the easiest part of PROTACs to modify in order to gain solubility, as the E3 ligase and POI binders are, to a certain extent, fixed.
6- Potential to use a universal ubiquitin ligase binder: It would be interesting to see your perspective on the possibility of using a universal ubiquitin ligase binder, including the potential advantages, disadvantages, and current evidence supporting this approach. A sentence is added on the notion of designing a universal ubiquitin binder (lines 155-159), noting that although it might simplify chemical synthesis in the first instance, the advantage of targeting specific tissues is a powerful advantage of non-ubiquitous e3 ligase binders.
Reviewer 2 Report
The Review by Thomas et al has the goal to summarize the literature on PROTAC technology towards CNS targets and, very importantly, to discuss the issues on applying such approach. The review is well written and the literature is broadly covered on the subject. Below, some suggestions and critical observations in order to improve the presentation of the review, as follows:
Major point:
The review should start by presenting a brief description of PROTAC technology, followed by the picture of the PROTAC technology regarding the CNS (by summarizing section 2) and, next the discussion on the limitations of such approach based on catalytic studies as presented in the Introduction section (page 2). A specific section discussing uptake of degraders to the CNS should be considered. Moreover, the review must be centered in the CNS. In conclusion: the topics might be reordered. Finally, a conclusion section summarizing the CNS PROTAC approach is missing.
Minor points:
Page 1, Keywords: no reason to repeat the word keyword
Figure 1: why not utilize same colors for POI identification (dark blue) and E3 (light blue)? The figure would be clearer to identify PROTAC elements.
Page 2, line 45 and 47 would not be a section and subsection, respectively?
Page2, line 53: PPis should be defined
Page2, line 56: cite is misspelling
Page 2: the “hook-effect” should be better explained. It is not clear as reported in fig 2 the bell-shaped curve. A simple question: why high PROTAC concentrations imply on decreased POI degradation?
Author Response
The review should start by presenting a brief description of PROTAC technology, followed by the picture of the PROTAC technology regarding the CNS (by summarizing section 2) and, next the discussion on the limitations of such approach based on catalytic studies as presented in the Introduction section (page 2). A specific section discussing uptake of degraders to the CNS should be considered. Moreover, the review must be centered in the CNS. In conclusion: the topics might be reordered. Finally, a conclusion section summarizing the CNS PROTAC approach is missing. A brief description, complete with a new figure, of PROTAC technology is included at the beginning. An image, depicting the topic is included in the abstract section and may be used as the graphic for the review. The subsections within the PROTAC sections have been rearranged, so that the mechanism for targeted protein degradation are described first (1.1), before moving on to other factors. Uptake to the CNS/bbb permeability is included as discussed above.
Reviewer 3 Report
In this review article by Simon E. ward et al., authors summarized advances in targeted protein degradation and its possible applications in addressing Central nervous system diseases.
Overall, the manuscript is well written with appropriate references. However, there is no new and interesting perspective the authors are bringing to the manuscript which s not reported in previous review articles that dealt with the subject of interest.
Comments:
1) Line 112,132 the error message is not clear what it is conveying.
2) Limitations of protac technology should be discussed further.
3) A table representing available technologies and their potential applications in each CNS pathologies can improve the overall readability of the manuscript.
Author Response
1) Line 112,132 the error message is not clear what it is conveying. Unclear what is meant by the reviewer. No error message is displayed for us.
2) Limitations of protac technology should be discussed further. The limitations of PROTACs have been included naturally as a response to reviewer 1's comments - the challenge of bbb and permeability in general is discussed, solubility of such large molecules and potential added complexity of dosing due to Hook's effect. Their disadvantages relative to smaller degrader molecules, are also discussed in the new section on molecular glues.
3) A table representing available technologies and their potential applications in each CNS pathologies can improve the overall readability of the manuscript. The most widely explored degrader technology, PROTACs, has a subsection for each CNS target. Although we agree that such a table would be beneficial in future, at present, we feel that there are only limited instances of the less explored emerging technologies, which would make the table quite bare. If the reviewer feels strongly, we are open to revisiting this point.
Round 2
Reviewer 2 Report
The present version of the manuscript sounds approptiate for publication.